# Ingestion of Nitrate and Nitrite and Risk of Stomach and Other Digestive System Cancers in the Iowa Women’s Health Study

**DOI:** 10.3390/ijerph18136822

**Published:** 2021-06-25

**Authors:** Ian D. Buller, Deven M. Patel, Peter J. Weyer, Anna Prizment, Rena R. Jones, Mary H. Ward

**Affiliations:** 1Occupational and Environmental Epidemiology Branch, Division of Cancer Epidemiology and Genetics, National Cancer Institute, Rockville, MD 20850, USA; ian.buller@nih.gov (I.D.B.); deven.m.patel.20@gmail.com (D.M.P.); rena.jones@nih.gov (R.R.J.); 2Cancer Prevention Fellowship Program, Division of Cancer Prevention, National Cancer Institute, Rockville, MD 20850, USA; 3Center for Health Effects of Environmental Contamination, University of Iowa, Iowa City, IA 52242, USA; peteweyer626@gmail.com; 4Masonic Cancer Center, Division of Hematology, Oncology and Transplantation, University of Minnesota, Minneapolis, MN 55455, USA; prizm001@umn.edu

**Keywords:** nitrate, nitrite, diet, drinking water contaminants, disinfection by-products, stomach cancer, esophagus cancer, small intestine cancer, liver cancer, gallbladder cancer

## Abstract

Nitrate and nitrite are precursors in the endogenous formation of N-nitroso compounds (NOC) which are potent animal carcinogens for the organs of the digestive system. We evaluated dietary intakes of nitrate and nitrite, as well as nitrate ingestion from drinking water (public drinking water supplies (PWS)), in relation to the incidence (1986–2014) of cancers of the esophagus (n = 36), stomach (n = 84), small intestine (n = 32), liver (n = 31), gallbladder (n = 66), and bile duct (n = 58) in the Iowa Women’s Health Study (42,000 women aged from 50 to 75 in 1986). Dietary nitrate and nitrite were estimated using a food frequency questionnaire and a database of nitrate and nitrite levels in foods. Historical nitrate measurements from PWS were linked to the enrollment address by duration. We used Cox regression to compute hazard ratios (HR) and 95% confidence intervals (CI) for exposure quartiles (Q), tertiles (T), or medians, depending on the number of cancer cases. In adjusted models, nitrite intake from processed meats was associated with an increased risk of stomach cancer (HR_Q4vsQ1_ = 2.2, CI: 1.2–4.3). A high intake of total dietary nitrite was inversely associated with gallbladder cancer (HR_Q4vsQ1_ = 0.3, CI: 0.1–0.96), driven by an inverse association with plant sources of nitrite (HR_Q4vsQ1_ = 0.3, CI: 0.1–0.9). Additionally, small intestine cancer was inversely associated with a high intake of animal nitrite (HR_T3vsT1_ = 0.2, CI: 0.1–0.7). There were no other dietary associations. Nitrate concentrations in PWS (average, years ≥ 1/2 the maximum contaminant level) were not associated with cancer incidence. Our findings for stomach cancer are consistent with prior dietary studies, and we are the first to evaluate nitrate and nitrite ingestion for certain gastrointestinal cancers.

## 1. Introduction

Cancers of the digestive system include tumors of the pancreas, liver, bile ducts, gallbladder, as well as the gastrointestinal (GI) tract, which includes the mouth, esophagus, stomach, small intestine, large intestine, and anus. The upper gastrointestinal tract includes the mouth, esophagus, and stomach. Four of the five cancer sites with the worst prognosis (esophagus, stomach, pancreas, liver, and lung) are GI cancers [1]. While the incidence of these cancers is rare (about 7% of all new cancer cases in the United States, combined), incidence rates differ by cancer types and have increased (small intestine, liver, and bile duct), decreased (stomach), or remained stable (esophagus and gallbladder) over the past decade in the United States [2,3]. Digestive system cancers occur primarily in older adults, with the median age of incidence from between 60 to 70 years old [2,3]. Risk factors for specific upper GI cancers include tobacco and heavy alcohol use for esophageal cancer [4]; *Helicobacter pylori* infection and alcohol use for stomach cancer [5,6]; hepatitis and aflatoxin for liver cancer [7]; and primary sclerosing cholangitis, cholelithiasis (gallstones), and liver flukes for gallbladder and bile duct cancers [8,9]. However, potential environmental risk factors for digestive system cancers have not been extensively investigated.

One potential risk factor for digestive system cancers that has not been widely evaluated is nitrate and nitrite intake. The International Agency for Research on Cancer (IARC) classified nitrate and nitrite as probable human carcinogens when ingested under conditions that result in endogenous nitrosation [10]. Endogenous nitrosation occurs when nitrate and nitrite ingested via diet or drinking water react with dietary amines or amides to form N-nitroso compounds (NOC) [11]. NOCs cause cancer at various sites in animals [10,11,12]. However, at the time of the IARC review, there was limited evidence for dietary nitrite carcinogenicity, which was based on studies of esophageal and stomach cancers, and inadequate evidence for ingested nitrate from drinking water [11].

The primary exposure to nitrate is typically through the ingestion of vegetables containing high levels of nitrate. Processed meats usually contain added nitrite or nitrate, resulting in the formation of carcinogenic NOC [13]. Red and processed meats also contain high heme iron levels, which increases NOC formation in the GI tract [14]. Drinking water can be a significant source in agricultural areas because of the high potential for nitrate contamination from nearby fertilizer applications and manure from animal farming [11,15,16]. The majority of the epidemiologic investigations of drinking water nitrate and digestive system cancers have focused on stomach cancer in areas where nitrate levels in the drinking water (from public or private sources) were elevated due to nearby agricultural practices [11].

Epidemiologic studies have assessed the relationship between ingested nitrates and nitrites and digestive cancers. A prospective study of nitrate and nitrite, estimated based on dietary intakes in the Netherlands, found that NOCs were positively associated with a risk for esophageal and stomach cancer subtypes [17]. A population-based case–control study in Nebraska, United States, found that the joint effect of a higher ingestion of drinking water nitrate and a higher processed meat intake versus a lower ingestion of drinking water nitrate and a lower processed meat intake was associated with a higher risk of stomach cancer but not esophageal cancer [18]. However, most of these studies did not quantify nitrate and nitrite in the diet, and they did not consider antioxidant intakes in their analysis [11]. While plants (i.e., vegetables) are the major dietary sources of nitrate, many of these sources also contain antioxidants (e.g., vitamin C) that inhibit NOC formation [19,20]. A prospective analysis of the Iowa Women’s Health Study (IWHS) by Weyer et al. [21] found a positive association between nitrate and the risk of digestive system cancers (grouped), including esophagus, stomach, small intestine, liver and bile ducts, gallbladder, peritoneum, and retroperitoneum.

There have been few studies of digestive system cancers and other water contaminants, including disinfection by-products (DBP), some of which are probable or possible human carcinogens [22,23]. Trihalomethanes are halogenated DBPs commonly found in chlorinated drinking water and have been positively associated with bladder cancer in epidemiologic studies but not with upper GI cancers [22,24,25]. Chlorinated DBPs are associated with tumor promotion in mice livers [26], and a halogenated hydroxyfuranone derivative promotes bile duct tumors in mice [27]. However, the few epidemiologic studies of DBP exposures and digestive cancers have not found associations. For example, two studies of esophageal cancer and DBP exposure found no significant associations [28]. An early investigation of chlorinated DBP in the IWHS did not find a relationship with upper GI cancer; however, upper GI cancer cases were rare and grouped [29]. To our knowledge, the association between DBP and many digestive system cancers has not been evaluated previously.

Our objective was to investigate the association between dietary and drinking water nitrate and nitrite ingestion and the risk of specific digestive system cancers (esophagus, stomach, small intestine, liver, gall bladder, bile ducts) among postmenopausal women in the IWHS. We extended the previous analysis of Weyer et al. [21] with an additional 13 years of follow-up. We also evaluated the relationship between DBP exposure and these cancers in the IWHS for the first time.

## 2. Materials and Methods

### 2.1. Study Population (IWHS)

As previously described, the Iowa Women’s Health Study is a large prospective cohort study of postmenopausal women in Iowa [30]. In 1986, 98,030 women aged from 55 to 69, who were randomly selected from the Iowa driver’s license record, were mailed a questionnaire that included demographics, medical and reproductive history, dietary intake, and family history of cancer. About 42% (N = 41,836) responded to the baseline questionnaire which formed the initial cohort. Participants completed five follow-up questionnaires (1987, 1989, 1992, 1997, 2004) with high response rates (91%, 90%, 83%, 79%, and 70%, respectively). The Institutional Review Boards of the University of Minnesota and the University of Iowa approved the IWHS.

Here, we analyzed specific digestive system cancers diagnosed from between 1986 and 2014 from the State Health Registry of Iowa, which included cancers (ICD-10 code group) of the esophagus (C15.x), stomach (C16.x), small intestine (C17.x), liver (C22.X), intrahepatic bile duct (C22.1), extrahepatic bile duct (C24.0), ampulla of Vater (C24.1), and gallbladder (C23.9). We excluded incident cases of lymphoid cancer, mesothelioma, and Kaposi sarcoma (N = 37) at these sites. Each participant’s vital status was obtained from the State Health Registry of Iowa and the National Death Index. We calculated person–years of follow-up from the enrollment date until the date of (1) incident cancer diagnosis, (2) death, (3) emigration from Iowa, or (4) the midpoint of the last contact date and 31 December 2014.

### 2.2. Exclusion Criteria

Using self-reported data at enrollment, we excluded participants who were premenopausal (N = 569) and who had been diagnosed with prior cancer (except non-melanoma skin cancer) or received cancer chemotherapy (N = 3830). We also excluded participants who reported unrealistic dietary intakes (<600 or >5000 kcal/day; N = 536), responded to 30 or fewer dietary questions (N = 2782) on the baseline survey or were missing key covariate information (N = 744). A total of 33,964 participants remained in the dietary analyses after these exclusions. An additional 4300 participants were excluded from the drinking water analysis because they had not participated in the 1989 follow-up survey on which drinking water source was reported. Of the participants who responded to the 1989 follow-up, we further excluded participants who reported using their water supply for ten years or fewer (N = 5600; 85% of cohort > 20 years), for an unreported period (N = 67), and those who reported drinking only bottled or other water sources (N = 1013). We excluded participants from municipalities for which we lacked sufficient public drinking water supply (PWS) source information, or had a groundwater aquifer or single surface water source for less than 75% of the study period (N = 2428), to reduce sources of uncertainty in the exposure assessment. We further excluded participants without NO_3_-N or total trihalomethanes (TTHM) measurements for their PWS (N = 4362). A total of 20,507 participants remained in the drinking water analyses after these exclusions (N = 13,457), with 15,577 participants on PWS and 4930 participants on private wells. As previously shown, excluded participants were similar to those retained in the drinking water analysis [31].

### 2.3. Exposure Assessment

#### 2.3.1. Diet

We assessed participant dietary intake at baseline using the Harvard food frequency questionnaire (FFQ) [32] which asks 127 questions about foods participants consumed in the past 12 months. We multiplied participant-reported intakes of food items by estimates of the nitrate and nitrite contents of the food as reported in the published literature [33,34] to estimate the total intakes of nitrate and nitrite from plant, animal, processed meat sources, and overall. Nitrate and nitrite intakes assessed using this database had comparable reliability to most macro- and micronutrients [35]. We also calculated red meat intake (g/day) and total vitamin C intake (mg/day) from foods and dietary supplements combined.

#### 2.3.2. Drinking Water

Based on the 1989 follow-up questionnaire (N = 36,127), we obtained the primary source drinking water at participants’ homes (municipal water system, rural water system, bottled water, private well water, other, do not know). The majority of participants (76.7%) indicated that they used a public water supply (rural water supply or municipal water system), 18.5% had a private well, and fewer than 5% reported using bottled or other water sources. Participants also reported the duration they had used this water source (years: ≤1, 1–5, 6–10, 11–20, ≥20). Ninety percent had used their drinking water source for longer than ten years.

We assessed contaminants in PWS similarly to previous investigations [21,36]. Briefly, we used nitrate-nitrogen (NO_3_-N) measured in water samples from municipal water supplies in Iowa across the 33-year historical exposure period (1955–1988) to calculate the annual average NO_3_-N (mg/L) concentrations for each PWS. We used the historical annual estimated concentration (µg/L) of TTHM, and the sum of the most prevalent DBP class, as a surrogate for the presence of other halogenated DBPs in our drinking water analyses. Experts estimated concentrations before the promulgation of regulations in the 1980s based on the known characteristics of the PWS, including treatment practices and water source measurements [37]. Using complete water source histories obtained from similarly aged women in Iowa during the same period [38], we estimated the median years within the reported drinking water source duration categories. Medians were 16 and 40 years for the 11–20 and >20 years’ categories, respectively. We based long-term NO_3_-N averages on the period of measurements available for each utility, with a maximum of 33 years, and we created two exposure metrics based on the duration of use: (1) long-term average based on years of use, and (2) the number of years in these periods when the annual average NO_3_-N or TTHM concentrations were greater than one-half of the maximum contaminant level (>1/2 MCL; NO_3_-N = 5 mg/L; TTHM = 40 µg/L).

### 2.4. Statistical Analysis

We used Cox proportional hazards regression to estimate hazard ratios (HR) and 95% confidence intervals (CI) of the association between dietary nitrate and nitrite, drinking water nitrate, and incident digestive system cancers. We grouped certain cancer types because we had small sample sizes for specific cancers. “Bile duct” cancers combined intrahepatic bile duct and extrahepatic bile duct, and ampulla of Vater and “biliary” cancers combined bile duct cancers and gallbladder cancer. In dietary analyses, we compared the risks in nitrate and nitrite intake quartiles and the 95th percentile to the risks in the lowest exposure quartile (Q1) for cancer groups with a sufficient total number of cases (N ≥ 50), including biliary, gallbladder, bile duct, and stomach cancers. We compared the risks in nitrate and nitrite intake tertiles and the 95th percentile to the risks in the lowest exposure tertile (T1) for esophagus, small intestine, and liver cancers. We tested for linear trends by including nitrate and nitrite intake as continuous variables. The intakes were log-transformed to achieve normality. We adjusted final dietary models for potential confounders assessed in the baseline questionnaire, including age and total calorie intake (kcal/day) as well as categorical smoking status (current smoker, former smoker, non-smoker), smoking pack-years, alcohol consumption (grams/day), body mass index (BMI), and categorical rurality (farm/rural area v. town/city) based on the lowest Akaike information criterion (AIC) during a stepwise model selection. We additionally examined the association between nitrite intake from various sources (plant, animal, and processed meat). For analyses of nitrite from animal sources and processed meats, we also adjusted models for saturated fat intake (g/day). We did not assess dietary nitrate from animal and plant sources separately because the majority of dietary nitrate (median proportion = 97%) was derived from plant sources [36].

For the drinking water analysis, we compared the risks in average nitrate and the TTHM exposure quartiles to the risk in the lowest exposure quartile (Q1) for biliary tract cancer (N ≥ 50) and exposure tertiles to the risk in the lowest exposure tertile (T1) for stomach cancer (50 < N ≥ 30). For esophagus, small intestine, gallbladder, bile duct, and liver cancers, we compared the risks in the above-median to the below-median (M1) exposure categories (N < 30). We evaluated participants with any years (>0 years) of exposure > 1/2 MCL compared to participants with no years (0 years) of exposure > 1/2 MCL. We transformed continuous variables using natural logarithms to achieve normality and tested for a linear trend by modeling continuous intakes. Because no nitrate data were available for participants using private wells, we compared them to participants on PWS in Q1 (or T1, M1) of the average nitrate level. We adjusted final drinking water models for age, as well as smoking status, smoking pack-years, alcohol intake, BMI, and categorical rurality based on the lowest Akaike information criterion (AIC) during an exhaustive model selection. We assessed effect modification by vitamin C (median split) and smoking status (ever/never) with stratified analyses for cancers with a sufficient frequency of cases (biliary and stomach cancers).

We used R (Version 3.6.3 [39]) and the *survival* package (Version 3.2-7 [40]) for all analyses, and we chose an alpha level of 0.05 for statistical significance.

## 3. Results

After exclusions, there were 308 digestive system cancer cases in the dietary analysis and 171 cases in the drinking water analysis. The mean participant age at the baseline was 61.6 years (SD = 4.2 years), and participants were followed up for an average of 21.6 years (SD = 8.1 years). Participants in the highest quartile of dietary nitrate intake were more often former smokers residing in towns with more than 1000 residents who had a lower median total caloric intake but a higher median intake of vitamins C and E than participants in the lowest quartile (Table 1). Participants on a PWS sourced from surface water varied across average nitrate categories, from a low of 5.2% in the 1st quartile to a high of 39.2% in the 4th quartile (Table 1). The median concentration of TTHM also varied across nitrate categories; there was a 9-fold increase from the 1st to the 4th quartile. Participants using a private well were more likely to live on a farm, smoke less or not at all, have a higher BMI, and consume more total calories but less vitamin C than those on PWS. We observed few other differences in demographic and lifestyle characteristics across the dietary nitrate and the drinking water nitrate categories.

We found no association between dietary nitrate and any digestive system cancers (Table 2). We found significant inverse associations between dietary nitrite and cancers (Table 2) of the biliary tract (gallbladder and bile ducts; HR_Q4vQ1_ = 0.37, CI: 0.16–0.85) and gallbladder (HR_Q4vQ1_ = 0.30, CI: 0.09–0.96), but a nonsignificant inverse association between dietary nitrite and cancer of the bile ducts. We found inverse relationships for both medium and high intakes of dietary nitrite and small intestine cancer, but only a statistically significant relationship for the medium level (HR_T2vT1_ = 0.36, CI: 0.14–0.94). We found nonsignificant associations between dietary nitrite and cancers of the stomach and esophagus and a nonsignificant positive association with liver cancer (HR_T3vT1_ = 3.37, CI: 0.93–12.21; Table 2). We did not find significant associations for continuous dietary intakes.

We found an inverse association between nitrite intake from plant sources and gallbladder cancer (HR_Q4vQ1_ = 0.32, CI: 0.12–0.86; HR_continuous_ = 0.49, CI: 0.24–0.98), but no association with any other digestive system cancer (Table 3). We found an inverse association between nitrite intake from animal sources and small intestine cancer (HR_T3vT1_ = 0.19, CI: 0.05–0.73; HR_continuous_ = 0.81, CI: 0.67–0.97). Nitrite intake from processed meats was inversely associated with esophageal cancer (HR_T2vT1_ = 0.17, CI: 0.06–0.51; HR_continuous_ = 0.90 CI: 0.83–0.97), but positively associated with stomach cancer risk (HR_Q4vQ1_ = 2.24, CI: 1.18–4.3; HR_continuous_ = 1.1, CI: 1.01–1.20).

In the drinking water analysis, we found no association between average nitrate concentrations and any of the digestive system cancers (Table 4). Similarly, we found no association between any cancer and years with exposure of >1/2 MCL of nitrate or with average nitrate as a continuous variable. We found no associations with private well water use compared to PWS participants in the lowest quartile of average nitrate for any cancer site (Table 4), although liver cancer risk was nonsignificantly elevated among private well users (HR = 2.6, CI: 0.7–10.3). We found no evidence of effect modification by vitamin C intake or smoking status for biliary and stomach cancers (Appendix A). Of the 14,939 participants on PWS with TTHM information, we found no association between any digestive system cancer and average TTHM quartiles, TTHM modeled continuously, or years with exposure > 1/2 MCL (Appendix A).

## 4. Discussion

Consistent with the literature on dietary nitrate intake and the risk of digestive system cancers, we found no associations between dietary nitrate intake and any of the digestive system cancers we evaluated. In contrast, we found inverse associations between total dietary nitrite intake and biliary tract and gallbladder cancers and a nonsignificant positive association with liver cancer. We observed weak inverse associations between various dietary nitrite intake sources and digestive system cancers, including plant nitrite and gallbladder cancer, animal nitrite and small intestine cancer, and processed meat nitrite and esophageal cancer. We did not observe significant relationships between drinking water nitrate or TTHM and digestive system cancers.

In prior analyses of dietary nitrate and digestive cancers in the IWHS that included cancers of the esophagus, stomach, small intestine, liver and bile ducts, gallbladder, peritoneum, and retroperitoneum, Weyer et al. [21] observed no significant association with dietary nitrate and these “other digestive” cancers (N = 71). Our updated analysis added 13 years of follow-up and 236 additional cancer cases (we did not include peritoneum cancers), for a total of 307. Quist et al. [41] assessed pancreas cancer in the IWHS and found no association with estimated nitrate ingestion from drinking water but found a positive association with dietary nitrite intake from processed meat. Jones et al. [42] assessed colon and rectum cancers in the IWHS and found no association with nitrate ingestion from drinking water but a positive association with red meat intake. We also conducted a more detailed dietary analysis by examining separate sources of nitrate and nitrite in relation to the risk of individual digestive system cancers. In 2010, the IARC concluded that there was neither enough evidence that ingested nitrate was an animal carcinogen nor that dietary nitrate was a human carcinogen [11]. Indeed, most dietary nitrate and digestive system cancer risk studies found no association or an inverse association [11], and we also found both inverse and no associations between dietary nitrate and the specific digestive system cancers we evaluated. A meta-analysis of epidemiologic studies of dietary nitrate and cancer found no significant associations for cancers of the esophagus (N = 4 studies) and stomach (N = 19) [43]. The same meta-analysis examined epidemiologic studies of dietary nitrite and found no associations for cancers of the esophagus (N = 7 studies) and stomach (n = 19) [43]. There were no studies of biliary tract, gallbladder, small intestine, or liver cancers included in the meta-analysis [43]; therefore, our analysis adds important epidemiological data for these sites.

Plant sources contain vitamin C, which inhibits nitrosation [19,20], and may explain the weakly protective association we observed between plant nitrite intake and gallbladder cancer. However, we saw no differences in nitrate associations across the strata of vitamin C intake for biliary tract or stomach cancers. We do not have an explanation for the inverse association between animal nitrite intake and small intestine cancer risk. Our results are not consistent with Cross et al. [44], who found that meat intake had a positive association with small intestine cancer risk in a larger cohort (NIH-AARP Diet and Health Study) that included men and women (results controlled for sex and did not stratify). In that study, participants were enrolled ten years after the IWHS and had higher dietary nitrate and nitrite intakes [35]. In 2010, the IARC concluded there was limited evidence for a positive association with dietary nitrite intake based on studies of stomach and esophageal cancers [11] and concluded there was limited evidence for a positive association between esophageal cancer and processed meat, a significant source of dietary nitrite [13]. We observed an inverse association between processed meat intake and esophageal cancer. This finding is inconsistent with the IARC monograph summary of the literature through 2006 that classified processed meat as a human carcinogen, which was primarily based on epidemiologic studies of colorectal cancer [13]. However, we observed a hazardous relationship between dietary nitrite from processed meat and stomach cancer, consistent with the IARC classification [13] and a meta-analysis of stomach cancer and processed meat intake [45]. Our findings were not consistent with those of Cross et al. [46] that found that dietary nitrate and nitrate intakes from processed meats were not associated with esophageal or stomach cancer risk. The IARC did not include studies of dietary nitrite and processed meat for biliary tract, gallbladder, or liver cancers [11,47]. For men and women, Nelson et al. [48] found a positive association for both preserved vegetables and salted meats and gallbladder cancers as well as similar relationships for biliary cancers such as extrahepatic bile duct and ampulla of Vater cancers.

Weyer et al. [21] found a suggestive positive association between drinking water nitrate and “other digestive” cancers (n = 55) within the IWHS. Our updated analysis added 72 digestive system cancer cases for a total of N = 127. We also improved the drinking water analysis by linking the water quality data by duration at the residence, evaluating TTHM, and evaluating the duration of exposure to levels > 1/2 MCL. We found no associations between nitrate and TTHM in drinking water and the risk of these digestive system cancers. The lack of evidence for an association between average nitrate exposure from drinking water and these cancers is consistent with the few other epidemiologic studies evaluating this relationship [49,50]. A prospective cohort study in the Netherlands, with lower average nitrate levels (1.31 mg/L as NO_3_-N) compared to the IWHS levels (1.84 mg/L), reported no association between drinking water nitrate and stomach cancer [51]. A population-based case–control study in Nebraska, United Stated, with higher median nitrate levels (2.58 mg/L) than the IWHS (1.08 mg/L), reported no associations between drinking water nitrate and stomach and esophagus cancers [18]. Similarly, a population-based case–control study in Yorkshire, United Kingdom, with higher median nitrate levels (1.83 mg/L) than in the IWHS, reported no associations between drinking water nitrate and stomach and esophagus cancers [52]. A small hospital-based case–control study in India observed a positive association between drinking water nitrate and stomach cancer but had higher median drinking water nitrate levels (4.6 mg/L) [53]. A population-based case–control study in Japan observed a positive association between drinking water nitrate and stomach cancer mortality at nitrate levels lower (median = 0.44 mg/L in controls) than the IWHS [54]. 

A small hospital-based case–control study in India of gallbladder cancer, using controls with cholelithiasis, measured the nitrate concentration in bile fluid and found a positive association between biliary nitrate concentration and gallbladder cancer [55], but the study was cross-sectional. To the best of our knowledge, no prior studies of biliary tract, liver, and small intestine cancers and drinking water nitrate have been conducted. Our study is also one of the first to examine the relationship between drinking water TTHM and these digestive system cancers. An early investigation of chlorinated DBP in the IWHS did not find a relationship with upper digestive system cancers as a group that excluded the lip and major salivary glands [29]. The same study found a significant association between chloroform levels (a trihalomethane) and colon cancer (n = 178) but not anus/rectum cancer (n = 78) [29]. A follow-up within the same cohort found no significant relationship between TTHM and colon cancer risk but significant positive associations between rectum cancer risk, TTHM, and specific DBPs [42]. A recent evaluation of TTHM concentrations and pancreatic cancer in the IWHS found no association with average exposure levels or with a duration of exposure to levels > 1/2 MCL [41]. We observed no significant relationships between TTHM and the specific digestive system cancers we evaluated. 

The main strength of our study is that we assessed exposure to nitrate and nitrite from both drinking water and from dietary sources for these rare digestive system cancers. There have been few studies, especially prospective cohort studies, that have evaluated both sources of these exposures. Other strengths include the availability of historical water quality data, a long period of follow-up, and low rates of residential mobility among the cohort participants [56].

Our analyses have some limitations. We had a small number of cases for some cancer sites, especially for the drinking water analyses. Small numbers precluded examining effect modification by vitamin C, red and processed meat intake, and smoking for the rarer digestive system cancers. The FFQ used to estimate dietary intake may be a source of exposure misclassification that is likely non-differential; however, a previous validation effort in the IWHS indicated good accuracy and reliability to support its usage to assess macronutrients [35,57]. We did not have information on the amount of the participants’ water consumption, which could have been used to estimate cumulative drinking water exposure. We excluded 18% of the IWHS participants with drinking water information because they were not on a water source for more than ten years, which allowed us to assess exposure periods of >20 years for 85% of the cohort. We were unable to assess other DBPs [58], pesticides [59], or heavy metal [60] exposures which may be risk factors for digestive system cancers. We did not have measurements of nitrate for participants using private wells. We also lacked information on well depth that can be used to estimate nitrate levels [61]. Private wells are typically not chlorinated and, therefore, we would expect few to no TTHM exposures. We did not have information on *H. pylori* infection, an important risk factor for stomach cancer [5,6].

## 5. Conclusions

We observed a positive association between a relatively low dietary intake of nitrite from processed meats and stomach cancer risk in postmenopausal women consistent with the majority of prior studies. We found inverse associations between dietary nitrite intake from plant sources and gallbladder cancer and nitrate from animal sources and small intestine cancers. However, these relationships were weak and based on a small number of cases. We found no association between long-term exposure to nitrate or TTHM levels in public water supplies and the risk of these digestive system cancers. This study adds important information on these exposures and digestive system cancer risks among women.

## Figures and Tables

**Table 1 ijerph-18-06822-t001:** Characteristics of Iowa Women’s Health Study participants in the dietary cohort (N = 33,964) and drinking water cohort (N = 20,507). The drinking water cohort included IWHS participants with >10 years at their drinking water source, by private well use (N = 4930), and nitrate-nitrogen (NO_3_-N) levels in public water (N = 15,577).

Characteristic	Dietary Intake *^a^* of NO_3_ (mg/day)				Private Well	Mean *^b^* NO_3_-N (mg/L) Levels in Public Water			
	<43.5	43.5–61.1	61.2–85.4	>85.4		<0.47	0.47–1.07	1.08–2.97	>2.97
Length of follow-up, years (mean ± SD)	21.4 ± 8.1	21.9 ± 8.1	21.8 ± 8.1	21.5 ± 8.3	23.3 ± 6.9	22.4 ± 7.2	22.2 ± 7.3	22.5 ± 7.2	22.4 ± 7.3
Age at baseline, years (mean ± SD)	61.4 ± 4.2	61.5 ± 4.2	61.7 ± 4.2	61.8 ± 4.1	61.2 ± 4.1	61.7 ± 4.2	61.6 ± 4.1	61.6 ± 4.2	61.6 ± 4.2
White race [%]	97.6	98.3	98.5	98.4	98.7	98.5	98.4	98.6	98.0
Surface water as source for PWS [%]	—	—	—	—	—	5.2	28.3	19.1	39.2
TTHM concentration, µg /L (median)	—	—	—	—	—	0.9	4.5	6.8	8.1
Nitrate in diet, mg/day (median) *^c^*	33.7	52.3	71.7	109.0	59.1	61.3	61.2	61.8	62.1
Nitrite in diet, mg/day (median) *^a^*	0.6	0.6	0.7	0.7	0.7	0.6	0.6	0.7	0.7
Plant nitrite in diet, mg/day (median) *^a^*	0.3	0.4	0.4	0.5	0.4	0.4	0.4	0.4	0.4
Animal nitrite in diet, mg/day (median) *^a^*	0.3	0.2	0.2	0.2	0.2	0.2	0.2	0.2	0.2
Processed meat nitrite in diet, µg/day (median) *^a^*	7.8	7.6	7.1	0.0	7.4	7.3	6.9	7.0	6.2
Vitamin C in diet, mg/day (median) *^a^*	80.5	98.5	111.5	142.4	99.8	107.9	111.5	109.7	111.4
Vitamin E in diet, mg/day (median) *^a^*	4.3	4.7	5.1	5.9	4.7	4.9	5.1	5.0	5.1
Saturated fat in diet, g/day (median) *^a^*	13.9	13.2	12.9	12.1	13.5	12.9	12.9	12.9	12.8
Total caloric intake, kcal/day (median)	1795	1775	1716	1597	1830	1698	1678	1690	1689
Alcohol intake (grams/day) [%]									
<14	91.5	92.2	92.1	92.2	95.2	90.7	91.1	90.7	90.4
≥14	8.5	7.8	7.9	7.8	4.8	9.3	8.9	9.3	9.6
Smoking status *^c^* [%]									
Never	64.9	67.5	67.6	64.8	78.9	63.1	62.1	61.3	61.8
Former	16.0	18.1	19.0	123.2	12.0	20.8	22.3	22.2	22.4
Current	19.3	14.3	13.5	12.0	9.1	16.0	15.5	16.4	15.8
Pack-years of smoking *^d^* [%]									
1–19	33.7	41.4	41.3	44.4	45.0	40.6	41.6	38.6	38.6
20–39	34.7	33.6	34.0	32.9	34.4	33.3	33.2	35.4	34.6
≥40	31.6	25.1	24.7	22.8	20.6	26.2	25.2	26.0	26.8
Residence [%]									
Farm	20.5	21.2	19.8	16.2	71.4	3.0	3.3	2.3	2.3
Rural area (nonfarm)	6.8	7.7	7.4	7.4	19.4	1.8	2.1	1.7	2.4
Towns ≥ 1000 residents	72.7	71.1	72.8	76.4	9.3	95.1	94.5	96.0	95.3
BMI (kg/m^2^) [%]									
<25	41.1	40.3	39.3	38.8	35.7	40.9	42.5	42.2	42.3
25–29.9	35.7	36.6	37.6	37.7	38.2	37.1	36.2	36.6	36.0
≥30	23.1	23.2	23.1	23.4	26.1	22.0	21.3	21.2	21.6

Abbreviations: BMI, body mass index; PWS, public drinking water supply; SD, standard deviation; TTHM, total trihalomethanes, *^a^* Adjusted for 1000 kcal per day of total energy intake. *^b^* Exposure assigned to individuals based on their reported duration at their drinking water source. *^c^* Determined based on most recent follow-up participation; otherwise from baseline report. *^d^* Among current or former smokers at baseline.

**Table 2 ijerph-18-06822-t002:** Association between dietary nitrate and nitrite (mg/day) and specific digestive system cancers in the Iowa Women’s Health Study (N = 33,964). *^a^* Quartile groups for biliary tract, gallbladder, bile duct, and stomach cancers. Tertile groups for esophagus, small intestine, and liver cancers.

	Dietary Nitrate (mg/day)					Dietary Nitrite (mg/day)				
Range	<71.7	71.8–106.1	106.2−151.5	>151.5	Continuous *^b^*	<0.9	0.9−1.1	1.1−1.4	>1.4	Continuous *^b^*
N	8466	8489	8506	8503		8533	8558	8583	8568	
**Biliary tract**										
Cases	35	35	31	23	124	42	33	31	19	124
HR *^c^* (CI)	ref	1.1 (0.7−1.7)	1.0 (0.6−1.6)	0.8 (0.4−1.4)	1.0 (0.7−1.3)	ref	0.7 (0.4−1.2)	0.7 (0.4−1.2)	0.4 (0.2−0.8)	0.7 (0.3−1.5)
**Gallbladder**										
Cases	23	18	12	13	66	24	18	15	9	66
HR *^d^* (CI)	ref	0.8 (0.4−1.5)	0.6 (0.3−1.1)	0.6 (0.3−1.3)	0.8 (0.5−1.2)	ref	0.7 (0.4−1.3)	0.6 (0.2−1.2)	0.3 (0.1−0.96)	0.5 (0.2−1.3)
**Bile Duct**										
Cases	12	17	19	10	58	18	15	16	9	58
HR *^c^* (CI)	ref	1.5 (0.7−3.2)	1.7 (0.8−3.6)	1.0 (0.4−2.4)	1.1 (0.7−1.9)	ref	0.8 (0.4−1.6)	0.8 (0.4−1.9)	0.4 (0.1−1.5)	1.2 (0.4−3.6)
**Stomach**										
Cases	23	24	20	17	84	17	24	23	20	84
HR *^e^* (CI)	ref	1.0 (0.5−1.7)	0.8 (0.4−1.4)	0.6 (0.3−1.2)	0.8 (0.5−1.2)	ref	1.2 (0.6−2.3)	1.1 (0.5−2.2)	0.8 (0.3−2.0)	1.4 (0.5−3.5)
** Range**	**<82.9**	**83.0−133.1**	**>133.2**	—		**<0.95**	**0.96−1.31**	**>1.31**	—	
** N**	**11,414**	**11,417**	**11,411**	—		**11,396**	**11,433**	**11,413**	—	
**Esophagus**										
Cases	12	14	10	—	36	16	11	9	—	36
HR *^f^* (CI)	ref	1.3 (0.6−2.8)	1.0 (0.4−2.5)	—	0.9 (0.5−1.8)	ref	0.9 (0.4−2.3)	1.1 (0.3−3.6)	—	1.5 (0.4−6.4)
**Small Intestine**										
Cases	13	12	7	—	32	15	7	10	—	32
HR *^g^* (CI)	ref	0.8 (0.4−1.9)	0.4 (0.2−1.2)	—	0.6 (0.3−1.1)	ref	0.4 (0.1−0.9)	0.4 (0.1−1.2)	—	0.3 (0.1−1.2)
**Liver**										
Cases	10	13	8	—	31	8	9	14	—	31
HR *^h^* (CI)	ref	1.3 (0.5−2.9)	0.8 (0.3−2.0)	—	0.7 (0.3−1.3)	ref	1.6 (0.6−4.6)	3.4 (0.9−12.2)	—	1.5 (0.3−6.9)

Abbreviations: BMI, body mass index; CI, 95% Confidence Interval; HR, Hazard Ratio, ^a^ Limited to those with non-missing covariate data. *^b^* Continuous variables for nitrate and nitrite were log-transformed. *^c^* Adjusted for age, calorie intake, and BMI. *^d^* Adjusted for age, calorie intake, and vitamin E intake. *^e^* Adjusted for age, calorie intake, and alcohol intake. *^f^* Adjusted for age, calorie intake, alcohol intake, and smoking pack-years. *^g^* Adjusted for age, calorie intake, and rurality. *^h^* Adjusted for age, calorie intake, BMI, and salt intake.

**Table 3 ijerph-18-06822-t003:** Association between dietary nitrite from plant, animal, and processed meat sources (mg/day) and specific digestive system cancers in the Iowa Women’s Health Study (N = 33,964). *^a^* Quartile groups for biliary tract, gallbladder, bile duct, and stomach cancers. Tertile groups for esophagus, small intestine, and liver cancers.

	Dietary Nitrite from Plant Sources (mg/day)					Dietary Nitrite from Animal Sources (mg/day) *^c^*				
** Range**	**<0.51**	**0.51–0.67**	**0.68–0.91**	**>0.91**	**Continuous *^b^***	**<0.29**	**0.29–0.40**	**0.41–0.57**	**>0.57**	**Continuous *^b^***
** N**	**8461**	**8507**	**8490**	**8506**		**8397**	**8520**	**8490**	**8557**	
**Biliary tract**										
Cases	39	32	31	22	124	35	37	27	25	124
HR *^d^* (CI)	ref	0.9 (0.5–1.4)	0.9 (0.5–1.5)	0.7 (0.4–1.3)	0.8 (0.5–1.4)	ref	1.2 (0.7–1.9)	0.9 (0.5–1.6)	1.0 (0.5–2.0)	0.9 (0.7–1.2)
**Gallbladder**										
Cases	25	18	14	9	66	17	18	17	14	66
HR *^e^* (CI)	ref	0.7 (0.4–1.3)	0.5 (0.3–1.1)	0.3 (0.1–0.9)	0.5 (0.2–0.97)	ref	1.3 (0.6–2.6)	1.4 (0.7–3.0)	1.5 (0.6–4.0)	1.4 (0.7–2.8)
**Bile Duct**										
Cases	14	14	17	13	58	18	19	10	11	58
HR *^d^* (CI)	ref	1.1 (0.5–2.4)	1.5 (0.7–3.3)	1.4 (0.5–3.7)	1.4 (0.7–3.1)	ref	1.1 (0.5–2.1)	0.6 (0.2–1.4)	0.7 (0.2–1.9)	0.9 (0.7–1.1)
**Stomach**										
Cases	18	28	19	19	84	20	19	18	27	84
HR *^f^* (CI)	ref	1.3 (0.7–2.5)	0.8 (0.4–1.6)	0.7 (0.3–1.6)	1.0 (0.5–1.8)	ref	0.9 (0.5–1.8)	0.9 (0.4–1.7)	1.3 (0.6–2.8)	1.1 (0.6–2.0)
** Range**	**<0.57**	**0.57–0.82**	**>0.82**	**—**		**<0.33**	**0.33–0.51**	**>0.51**	**—**	
** N**	**11,281**	**11,327**	**11,356**	**—**		**11,242**	**11,322**	**11,400**	**—**	
**Esophagus**										
Cases	15	13	8	—	36	13	9	14	—	36
HR *^g^* (CI)	ref	1.1 (0.5–2.5)	0.9 (0.3–2.6)	—	1.1 (0.4–2.8)	ref	1.0 (0.4–2.4)	2.2 (0.8–6.2)	—	1.1 (0.5–2.7)
**Small Intestine**										
Cases	11	12	9	—	32	12	15	5	—	32
HR *^h^* (CI)	ref	1.0 (0.4–2.3)	0.6 (0.2–1.9)	—	0.9 (0.3–2.4)	ref	0.9 (0.4–2.1)	0.2 (0.1–0.7)	—	0.8 (0.7–0.96)
**Liver**										
Cases	12	8	11	—	31	9	8	14	—	31
Cases	ref	0.6 (0.2–1.6)	0.8 (0.3–2.4)	—	0.8 0.3–2.2)	ref	1.1 (0.4–3.0)	2.2 (0.7–6.9)	—	2.0 (0.7–5.7)
		**Dietary Nitrite from Processed Meat (mg/day) *^b^***								
** Range**		**<0.005**		**0.005–0.033**		**0.034–0.065**		**>0.065**		**Continuous *^c^***
** N**		**8718**		**8579**		**8246**		**8421**		
**Biliary tract**										
Cases		31		32		30		31		124
HR (CI) *^d^*		ref		1.1 (0.7–1.8)		1.1 (0.7–1.9)		1.4 (0.8–2.3)		1.0 (1.0–1.1)
**Gallbladder**										
Cases		18		16		16		16		66
HR (CI) *^e^*		ref		1.0 (0.5–1.9)		1.1 (0.5–2.1)		1.2 (0.6–2.5)		1.0 (0.9–1.1)
**Bile Duct**										
Cases		13		16		14		15		58
HR (CI) *^d^*		ref		1.4 (0.6–2.8)		1.3 (0.6–2.8)		1.6 (0.7–3.6)		1.1 (1.0–1.2)
**Stomach**										
Cases		16		18		18		32		84
HR (CI) *^f^*		ref		1.2 (0.6–2.3)		1.2 (0.6–2.4)		2.2 (1.2–4.3)		1.1 (1.01–1.2)
** Range**		**<0.02**		**0.02–0.05**		**>0.05**		—		
** N**		**10,067**		**12,190**		**11,707**		—		
**Esophagus**										
Cases		19		4		13		—		36
HR (CI) ***^g^***		ref		0.2 (0.1–0.5)		0.6 (0.3–1.4)		—		0.9 (0.8–0.99)
**Small Intestine**										
Cases		11		13		8		—		32
HR (CI) *^h^*		ref		0.9 (0.4–2.1)		0.6 (0.2–1.5)		—		1.0 (0.9–1.1)
**Liver**										
Cases		7		12		12		—		31
HR (CI) *^i^*		ref		1.4 (0.5–3.6)		1.5 (0.5–4.0)		—		1.0 (0.9–1.1)

Abbreviations: BMI, body mass index; CI, 95% Confidence Interval; HR, Hazard Ratio, *^a^* Limited to those with non-missing covariate data. *^b^* Adjusted for saturated fat intake. *^c^* Continuous variables for nitrate and nitrite were log-transformed. *^d^* Adjusted for age, calorie intake, and BMI. *^e^* Adjusted for age, calorie intake, and median vitamin E intake. *^f^* Adjusted for age, calorie intake, and alcohol intake. *^g^* Adjusted for age, calorie intake, smoking pack-years, and alcohol intake. *^h^* Adjusted for age, calorie intake, and rurality. *^i^* Adjusted for age, calorie intake, BMI, and salt.

**Table 4 ijerph-18-06822-t004:** Association between specific digestive system cancers and average drinking water nitrate-nitrogen (mg/L), continuous nitrate (log nitrate), years above half the maximum contaminant level (MCL), and private water (N = 20,507). *^a^* Quartile groups for biliary tract cancer. Tertile groups for stomach cancer. Median groups for esophagus, small intestine, gallbladder, bile duct, and liver cancers.

	Private *^b^*	Average NO_3_-N (mg/L)				Continuous (log Nitrate)	Years with > ½-MCL (>5 mg/L NO_3_-N)	
** Range**	**—**	**<0.47**	**0.48–1.07**	**1.08–2.97**	**>2.97**		**0 years**	**>0 years**
** N**	4930	3977	3724	3617	4259	15,710	10,947	4630
**Biliary tract**								
Cases	16	12	12	9	17	50	33	17
HR (CI) *^c^*	1.1 (0.5–2.3)	ref	1.1 (0.5–2.4)	0.8 (0.4–2.0)	1.3 (0.6–2.8)	1.2 (0.9–1.5)	ref	1.2 (0.7–2.2)
** Range**	**—**	**<0.61**	**0.61–2.36**	**>2.36**	—			
** N**	4930	5205	5182	5190	—			
**Stomach**								
Cases	12	10	10	10	—	30	23	7
HR (CI) *^c^*	1.3 (0.5–2.9)	ref	1.0 (0.4–2.4)	1.0 (0.4–2.5)	—	1.0 (0.7–1.4)	ref	0.7 (0.3–1.7)
** Range**	**—**	**≤** **1.08**	**>1.08**	**—**	**—**			
** N**	4930	7790	7787	**—**	**—**			
**Esophagus**								
Cases	4	9	12	—	—	21	12	9
HR (CI) *^d^*	0.8 (0.2–2.6)	ref	1.3 (0.5–3.1)	—	—	1.0 (0.7–1.5)	ref	1.8 (0.8–4.2)
**Small Intestine**								
Cases	5	11	4	—	—	15	12	3
HR (CI) *^e^*	0.7 (0.1–6.4)	ref	0.4 (0.1–1.1)	—	—	0.8 (0.5–1.3)	ref	0.6 (0.2–2.1)
**Gallbladder**								
Cases	9	13	12	—	—	25	18	7
HR (CI) *^e^*	0.8 (0.1–4.9)	ref	0.9 (0.4–2.0)	—	—	1.1 (0.8–1.6)	ref	0.9 (0.4–2.2)
**Bile Duct**								
Cases	7	11	14	—	—	25	15	10
HR (CI) *^c^*	1.3 (0.4–4.0)	ref	1.3 (0.6–2.8)	—	—	1.2 (0.8–1.8)	ref	1.6 (0.7–3.5)
**Liver**								
Cases	7	4	7	—	—	11	7	4
HR (CI) *^f^*	2.6 (0.7–10.3)	ref	1.8 (0.5–6.1)	—	—	1.0 (0.6–1.8)	ref	1.4 (0.4–4.6)

Abbreviations: BMI, body mass index; CI, 95% Confidence Interval; HR, Hazard Ratio, ^*a*^ Limited to women 11+ years on PWS, and those with non-missing smoking status. *^b^* Compared to lowest group of average NO3-N on PWS. *^c^* Adjusted for age and BMI. *^d^* Adjusted for age, smoking status, and alcohol intake. *^e^* Adjusted for age and rurality. *^f^* Adjusted for age, BMI, and smoking pack-years.

## Data Availability

The data presented in this study are available on request from the corresponding author. The data are not publicly available due to privacy.

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
