# Peer review of "Ingestion of Nitrate and Nitrite and Risk of Stomach and Other Digestive System Cancers in the Iowa Women’s Health Study"

_ijerph, 2021, doi:10.3390/ijerph18136822_

Round 1

Reviewer 1 Report

Authors aim to evaluate dietary intakes of nitrate and nitrite as well as nitrate ingestion from drinking water (public drinking water supplies [PWS]) in relation to incidence (through 2014) of cancers of the esophagus (n=36), stomach (84), small intestine (32), liver (31), gallbladder (66), and bile duct (58) in the Iowa Women’s Health Study (42,000 women ages 50-75 in 1986) from the previous analysis of Weyer et al. with an additional 13 years of follow-up. Dietary nitrate and nitrite were estimated using a food frequency questionnaire and a database of nitrate and nitrite levels in foods. In adjusted models, nitrite intake from processed meats was associated with an increased risk of stomach cancer (HRQ4vsQ1=2.2, CI:1.2-4.3). High intake of total dietary nitrite was inversely associated with gallbladder cancer (HRQ4vsQ1=0.3, 24 CI:0.1-0.96), driven by an inverse association with plant sources of nitrite (HRQ4vsQ1=0.3, CI:0.1-0.9). Also, small intestine cancer was inversely associated with a high intake of animal nitrite 26 (HRT3vsT1=0.2, CI:0.1-0.7). In summary, the present study indicated that nitrate concentrations in PWS (average, years ≥1/2 the maximum contaminant level) were not associated with cancer incidence. Their stomach cancer findings are consistent with prior dietary studies and are first to evaluate nitrate/nitrite ingestion for certain gastrointestinal cancers. The results seems informative and appealing; the manuscript is well-written with powerful scientific findings.

Major Compulsory Revisions:     

  1. The major objective was to investigate the association between dietary and drinking water nitrate and nitrite ingestion and risk of specific digestive system cancers (esophagus, stomach, small intestine, liver, gall bladder, bile ducts) among postmenopausal women in the IWHS. Obviously, their study have shown the important results of dietary nitrite intake was associated with an increased risk of stomach cancer. Upmost important, nitrate concentrations in PWS were not associated with cancer incidence. The current study is well-designed, well-organized and well written to have a large cohort to show the vital public health information.
  2. 3.2. Drinking water. Participants also reported the duration they had used this water source (years: ≤1, 1–5, 6–10, 11– 20, ≥20). Ninety percent had used their drinking water 161 source for longer than 10 years. Is any difference among various time period exposure that is associated with GI cancer? In 2.3.1. Diet, no different time period exposure in diet?
  3. The first concern is how they stratify different dietary intake of NO3 (mg/day) dose, and mean NO3-N (mg/L) levels, dietary nitrate (mg/day) and dietary nitrite (mg/day) subgroups at various cut-off value?
  4. In Table 4. Years with >½-MCL (>5 mg/L NO3-N) parameter was divided into o years vs > 0 years?

Reviewer 2 Report

In the manuscript, Ian D Buller et al, determined the correlation of digestive cancer with nitrate and nitrite consumption. The text is generally well written and the data supports the conclusion. The reviewer have few minor suggestions that could improve the manuscript.

  • The Introduction sections are too long and should be short and more informative.
  • Same as applicable to discussion part as well.

1. Methods, provide a flow chart that summarizes the clinical design and data analysis  

2. Providing a simple representative bar diagram is more easy and effective for the readers.

3. Helicobacter is one of the important factors in stomach and digestive system cancer. In the introduction, the author includes the relationship between ingestion of nitrate, nitrite on H.pylori and its various strains stomach and digestive cancer.

4. It is better to include in the results section also if they have any H. pylori related data relevant to nitrite and nitrate. 

Reviewer 3 Report

Dear Authors,

The paper itself is very well written and very interesting. Here are several minor suggestions:

  • I would recommend modifying the title of the article since the one proposed by you is quite not appealing. Maybe using such phrases as 'cancers of the gastrointestinal tract' instead of 'stomach and other digestive system cancers' would be more appealing? Please consider this.
  • Lines 16-17 - please add 'n=' in all of the brackets (not only in the first one.
  • Line 16 - do I understand it properly, is data evaluated in this manuscript only data collected in 2014? Or in the range of 2014-2021? In line 110 is is more precise. Please precise this in line 16 as well to be more clear.
  • Please check the guidelines for Authors (tables should be mentioned just below where they are mentioned in the text).
  • Please check the manuscript once again in terms of English.

Good luck with your further research

Best regards

A Reviewer
